

# Strain heterogeneities at the ductile to brittle transition; a case study on ice.

Chauve Thomas[1], Montagnat Maurine[1], Lachaud Cedric[1], Georges David[1], and Vacher Pierre[2]

[1]IGE, Université Grenoble Alpes / CNRS F-38041 Grenoble, France
[2]Laboratoire SYMME, Université de Savoie Mont Blanc, BP 80439, 74944 Annecy le Vieux Cedex, France

*Correspondence to:* maurine.montagnat@univ-grenoble-alpes.fr

**Abstract.** This paper presents, for the first time, the evolution of local strain fields around intragranular cracking in polycrystalline ice, at the onset of tertiary creep. Owing to the high homologous temperature conditions and relatively low compressive stress applied, stress concentration at crack tips is relaxed by plastic mechanisms associated with dynamic recrystallization. Strain field evolution followed by Digital Image Correlation indirectly shows the redistribution of stresses during crack opening, but also driven by crack tip plasticity mechanisms and recrystallization. Such redistribution induces modifications in the local strain deformation bands, and crack closure during deformation. A strong interaction between cracking and dynamic recrystallization is therefore evidenced at the ductile to brittle transition in ice deformed at high homologous temperature.

## 1 Introduction

The evaluation and the characterization of strain heterogeneities is of primary importance in material sciences at various scales of observation. Plastic strain localization in metals play a crucial role on the propagation of fracture and on the response to fatigue conditions, and Portevin-Le-Chatelier is a strong example of strain heterogeneities development during mechanical tests in some metal alloys (see Antolovich and Armstrong (2014) for a review). Similarly, strain heterogeneities and localization are known to strongly influence the rheological behavior of the Earth lithosphere. Although processes to explain strain localization remain unclear, coupling between deformation in frictional faults in the uppermost crust and localized shearing in the ductile crust and mantle seems to be necessary to explain post-seismic deformation (Tommasi et al., 2009; Vauchez et al., 2012).

In the contest of ice sheet flow, successive layers of ice with slightly different viscosity can experience different strain history as a result of strain localization initiated by bedrock topography (Paterson, 1994; Durand et al., 2007). Strain localization can induce flow disturbances that can mix the climatic signal and counteract the search for the oldest ice (Dahl-Jensen et al., 2013; Fischer et al., 2013). These flow disturbances can form as folding, that is observed at large scale from ice-penetrating radar surveys now able to highlight deep stratigraphy (MacGregor et al., 2015; Panton and Karlsson, 2015; Bons et al., 2016), but also at smaller scales from microstructure observations (Jansen et al., 2016).

During laboratory deformation experiments, strain heterogeneities can result from imposed boundary conditions. But in the mean time, the local stress field is strongly influenced by crystal plasticity anisotropy coupled with the microstructure. The resulting strain field can therefore depart from the main imposed stress patterns, and become strongly heterogeneous. This





was shown in halite, for instance, where Bourcier et al. (2013) were able to observe in-situ strain field associated with crystal plasticity and grain boundary sliding, as a response to the complex stress pattern. In turn, the role of ice microstructure correlated with its crystalline properties on strain heterogeneities were revealed recently by Digital Image Correlation measurements (Grennerat et al., 2012; Chauve et al., 2015).

During ductile deformation in natural or laboratory conditions (at high homologous temperature $\sim 0.97$ $T_m$, low strain rate $\sim 10^{-7} s^{-1}$ and low stress, 0.5 - 1 MPa), plastic deformation in ice is mainly accommodated by the glide of basal dislocations (Duval et al., 1983). The resulting strongly anisotropic viscoplastic behavior of the single crystal (Duval et al., 1983) leads to the development of strong strain heterogeneities during deformation of polycrystalline ice.

Strain heterogeneities evaluated during transient creep of ice, were shown to reach local values higher than 10 times the macro-
scopic strain, and to settle into bands which dimensions are higher than the grain size. Strain localization bands may follow grain boundaries, but also cross entire grains, and there is no statistical link between the crystallographic orientation and the amount of local strain (Grennerat et al., 2012). These first measurements of strain localization during laboratory experiments were restricted to transient (or primary) creep conditions, in ductile conditions ($\sigma < 0.5$ MPa and T $> 0.97$ $T_m$) and prior to any microstructure modification due to dynamic recrystallization.

More generally, creep of isotropic polycrystalline ice is characterized by a three-stages behavior, with a strong decrease of strain-rate during primary creep, down to a minimum reached at about 1% strain, also called secondary creep, immediately followed by a increase in strain-rate to reach tertiary creep at about 10% strain (see Jacka and Maccagnan (1984); Duval et al. (1983) for instance). At the onset of tertiary creep, dynamic recrystallization mechanisms will occur increasingly to relax the kinematic hardening and enable for further ductile deformation to occur (Duval et al., 1983). Dynamic recrystallization leads to
strong modification in microstructure and texture (Duval, 1979; Jacka and Maccagnan, 1984; Montagnat et al., 2015). Recently, nucleation mechanisms at the origin of such modification could be characterized in details thanks to Electron BackScattering Diffraction (EBSD) (Chauve et al., 2017), and were shown to be closely related to strain localization (Chauve et al., 2015). In particular, strain heterogeneities seem to induce a large variety of nucleation mechanisms, from polygonisation associated with subgrain boundary formation, to bulging resulting from Strain Induced Boundary Migration (SIBM) (Chauve et al., 2017).

While Piazolo et al. (2015) showed that sub-grain boundary formation such as kink bands could be correlated with heterogeneities of local stress (simulated with a full-field crystal plasticity code, CraFT), Chauve et al. (2015) was able to directly associate nucleation mechanisms (polygonisation, bulging) to local modification of the strain field estimated in-situ from DIC measurements.

Increase in strain-rate after secondary creep can also be associated with the occurrence of microcracking without a total collapse
of the sample. Such a configuration occurs at higher imposed stress (typically above 0.9 MPa) or higher imposed strain-rate ($\dot{\varepsilon} > 8.10^{-5} s^{-1}$) at temperature around -10°C (Schulson et al., 1984; Batto and Schulson, 1993; Schulson and Duval, 2009). The local stress field is therefore relaxed by cracks opening at or close to grain boundaries, and depending on the boundary conditions, crack propagation can occur at various rate. This mechanical response is typical of a ductile to brittle transition (Schulson and Buck, 1995; Schulson and Duval, 2009).

In this domain, most of the studies performed so far, some of which mentioned here, concentrated on macroscopic parameters



(deformation and creep curves, evaluation of the effect of temperature and grain size on the strength) and optical observations of the full sample to characterize the nature of the cracks, and of the fractures (Batto and Schulson, 1993; Iliescu and Schulson, 2004). From these observations, a theoretical framework was elaborated based on the assumption of the formation of wing cracks at the tip of initial cracks to relax the local stress field (Renshaw and Schulson, 2001). In particular, the conditions re-

quired to form these secondary cracks were shown to control the ductile to brittle transition under compression. More recently, Snyder et al. (2016) showed that this model was able to take into account the effect of a prestrain, including recrystallization mechanisms, on the increase of ductile-to-brittle transition strain-rate for ice.

At the ductile to brittle transition, mixture of creep by dislocations and cracking will occur, and it is related to the ability of the material to relax the stress accumulated at the tip of the initial cracks. For instance, Batto and Schulson (1993) showed that a

small amount of creep relaxation at the crack tip could be enough to postponed the transition to brittle behavior (in time or in strain-rate level). The mechanism of relaxation of the stress produced by a crack opening in mode I, through rapid multiplication of dislocations at crack tip has been reviewed by Argon (2001) for metallic materials. More recently, Martínez-Pañeda and Niordson (2016) was able to simulate the complexity of the effet of strain gradient plasticity on the level of stress at crack tip and on crack-tip blunting. Crack-tip-initiated plasticity is an crucial mechanism to explain a ductile-like behavior at the ductile

to brittle transition.

Although local stress field is hardly accessible experimentally, it can be indirectly and qualitatively approached from strain field evaluation at a local scale. In the present work we use Digital Image Correlation (DIC) technique, already well approved on ice, to evaluate the strain field evolution during a creep experiment on ice polycrystal performed at the ductile to brittle transition. After a brief presentation of the experimental set-up (Part 2), Part 3 will explore stress conditions during which

strain-rate increase with tertiary creep results from local cracking. We will see that plasticity is strongly active at crack tips as evidenced by the occurrence of dynamic recrystallization mechanisms. These mechanisms indeed play a crucial role to reduce and redistribute the local stress concentration that appears at the crack tips during the ductile to brittle transition.

## 2 Experimental set-up

Unconfined uniaxial creep tests have been carried out on polycrystalline columnar ice samples of type $S2$ (Ple and Meyssonnier, 1997). Parallelipedic samples ($\sim 90 \times 90 \times 15$ mm$^3$) were built and the column axes were positioned perpendicularly to the larger surface, and to the compression axis (figure 1). By doing so, the samples provide a "2D-1/2" microstructure, from which surface characterization can approximate volume behavior. Sample microstructure and texture were measured using an Automatic Ice Texture Analyser (AITA) (Wilson et al., 2003; Peternell et al., 2011), which is an optical technique measuring

the **c**-axis (or optical axis) orientation (azimuth $\theta$ and colatitude $\phi$) with a spatial resolution from 50 to 5 $\mu$m, and an angular resolution of about 3°. Although large areas can be analysed (up to $120 \times 120$ mm$^2$), this technique requires the preparation of thin sections of ice ($\sim 0.3$ mm thick), and is then destructive. By taking advantage of the columnar microstructure, we were able to compare *pre-* and *post-* deformation microstructures by carefully extracting thin layers of ice on top of the sample,





before and after the test (figure 1). Details of the procedure for sample preparation can be found in (Grennerat et al., 2012; Chauve et al., 2015).

During the experiment, DIC analyses were performed over the full surface of the samples by following the procedure adapted to ice by Grennerat et al. (2012). DIC provides in-situ measurements of the displacement and therefore strain field on the sample

surface, from the correlation of surface images of a grey-level speckle supposed to follow the sample deformation. By taking advantage of the 2D-1/2 configuration, we assumed the surface strain field to correctly represent the volume deformation, making possible the comparison between the measured microstructure by AITA and the strain field evaluated by DIC (figure 1).

The spatial resolution strongly depends on the quality of the speckle, the illumination and the sensitivity of the camera used.

In the following experiments, we used a Phase One 80 Mpx camera, the speckle was made of shoe polish that offers a good cohesion with the ice surface, and a good illumination was obtained thanks to two neon lamps. From that, we ended up with a spatial resolution of 0.19 mm.pix$^{-1}$, and a strain resolution between $3.10^{-3}$ and $4.10^{-3}$ for the different strain components (see table 2).

Displacement and strain data were extracted using the $7D$ software from Vacher et al. (1999). This DIC method provides

| Camera | DIC spatial resolution | DIC strain resolution | | |
|---|---|---|---|---|
| | | $\sigma_{\varepsilon_{xx}}$ | $\sigma_{\varepsilon_{yy}}$ | $\sigma_{\varepsilon_{xy}}$ |
| Phase One 80 Mpx | $0.19\,mm.pix^{-1}$ | $4.10^{-3}$ | $3.10^{-3}$ | $4.10^{-3}$ |

**Table 1.** Characteristics of the DIC measurements.

a set of displacement vectors over a given grid, defined for the DIC calculation as a function of the speckle and picture qualities (Vacher et al., 1999). From the displacement field components, the strain components are extracted by using Green-Lagrange expression. In-plane components of strain are therefore provided ($\varepsilon_{xx}$, $\varepsilon_{yy}$, and $\varepsilon_{xy}$), from which an equivalent strain ($\varepsilon_{eq} = \sqrt{\frac{2}{3}\left(\varepsilon_{xx}^2 + \varepsilon_{yy}^2 + 2\varepsilon_{xy}^2\right)}$) and principal strain components are calculated. The later will be plotted along their principal directions in the following figures.

Discontinuities such as cracks produce displacements which translation in terms of strain is not direct but could be estimated as shown by Nguyen et al. (2011). In the present study, we will simply use the direction of the principal strain components calculated around a crack to interpret the direction of the crack opening (or closing), since the displacement produced is small enough to be followed by the speckle on each side of the crack.

Since all surfaces except the loaded ones remained free (unconfined tests), a slight amount of out-of-plane shear cannot be

excluded. The effect of a deformation going out of the plane $xOy$ was estimated in previous analyses performed by Grennerat et al. (2012) and shown to remain low, in the limit of the small macroscopic deformations reached in the present study (less than 5.5%). In order to reduce the noise and this out of plane strain effect on the evaluation of the strain evolution during the experiment, we calculated the strain field during short increments of macroscopic deformation of 0.1% to 0.5%. Besides, observation of the incremental strain field enables to individualise consecutive events that would be hidden in a strain field




calculation integrating the whole experiment duration.

Table 2 summarizes the experimental conditions of the test that will be used as an illustration in this paper, and figure 2

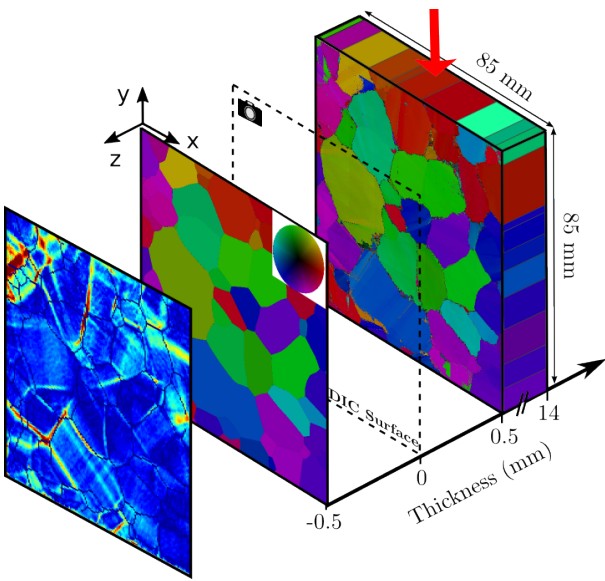

**Figure 1.** Scheme of the experimental set-up showing the shape of the sample with the direction of the imposed stress (red arrow). The position 0 corresponds to the sample surface (during the test) on top of which the speckle is marked. The microstructure prior to deformation analysed by AITA is located at about $-0.5$ $mm$ and the one after deformation is at about $0.5$ $mm$ from the sample surface ($0.5$ $mm$ corresponds to the ice thickness needed to make the thin section).

provides the creep curves. The minimum strain rate is reached at about 0.5% of compressive macro strain, slightly before the classical 1% value. This can be attributed to a microstructure effect since our 2D-1/2 samples contain only few grains and do not consist of good Representative Volume Elements. In the following, a negative sign will be given to the compressive strain, at the macroscopic and local scale.

| Stress (MPa) | Temperature (°C) | Strain rate $(s^{-1})$ | |
|---|---|---|---|
| | | mini ($\varepsilon_{yy} = -0.5\%$) | end ($\varepsilon_{yy} - 5.5\%$) |
| 1.0 | $-7$ | $5.0 \times 10^{-6}$ | $8.1 \times 10^{-5}$ |

**Table 2.** Experimental conditions at the ductile to brittle transition for the illustrative test. The minimum creep rate is reached at about -0.5% strain, slightly before the classical 1%.

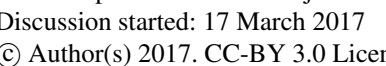



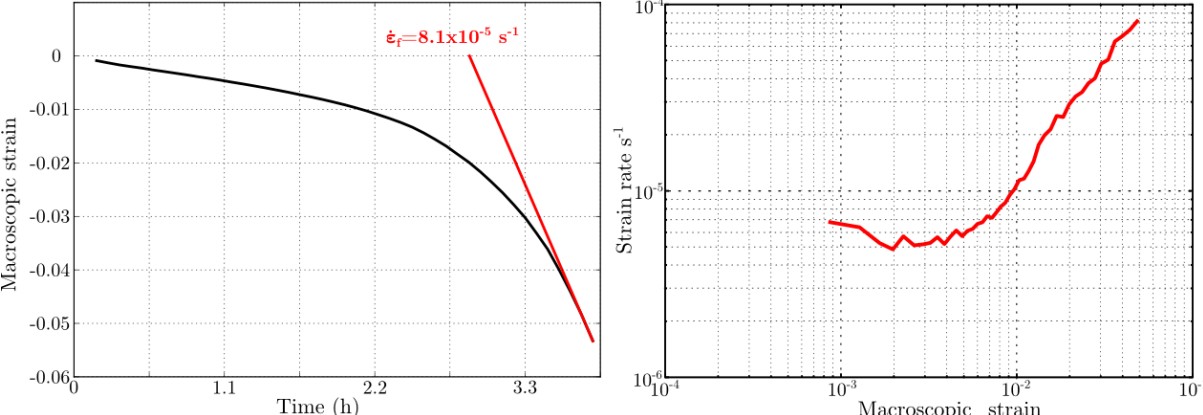

**Figure 2.** Evolution of the macroscopic strain and strain-rate measured from DIC calculations. Values before $10^{-3}$ macro strain where not calculated.

## 3 Strain field evolution at the ductile to brittle transition

The macroscopic strain curve reveals an increase in strain rate after -0.5% of $\varepsilon_{yy}$ (vertical) macro strain (figure 2). While at -0.5% of macro strain the minimum strain-rate was $5.0 \times 10^{-6}\ s^{-1}$, at the end of the experiment the strain-rate reached $8.1 \times 10^{-5}\ s^{-1}$, evidencing a strong acceleration at the onset of tertiary creep captured here.

The initial microstructure of the sample, the finale one, and an optical observation of the sample at the end of the test are shown in figure 3. Thanks to the transparency of ice, cracks and de-cohesion features can be observed with natural light. They appear as grey and black areas in figure 3. From the c-axis orientation color-scale, one can see that the intial texture is not isotropic. On top of the expected columnar grain-shape effect, we therefore expect a macroscopic mechanical response different from the one of an isotropic granular sample.

The global strain field measured prior to any visible crack opening on the speckle, at $-0.5\%$ of macro strain (at the minimum creep rate) is represented in figure 4 via the equivalent strain $\varepsilon_{eq}$ at two different spatial resolutions in order to illustrate the structure of strain heterogeneities. Similarly to what was already observed by Grennerat et al. (2012), the deformation is organised into bands crossing most of the sample. The main orientation of the bands is about 20 to 30° from the compression direction. Local equivalent strain amplitude in the deformation bands can reach more than 10%, for a $\varepsilon_{yy}$ macro strain of about 0.5%.

In the following a focus will be given on a small area located within the dashed black rectangle of figure 3. The initial microstructure and orientations of the grains in this area, the final microstructure where cracks, subgrain boundaries, and small nucleated grains appear, and a picture of the speckle from the surface of the sample where crack locations are visible (arrows 1 to 4) are shown in figure 5. Very small grains visible in the cracks are artifacts from the thin sectioning process (shaving

produces small ships that fill the crack interior), but new grains from recrystallization mechanisms can be distinguished away from the crack interior. Grain boundaries surrounding new small grains also appear perturbed because of intrinsic limitation





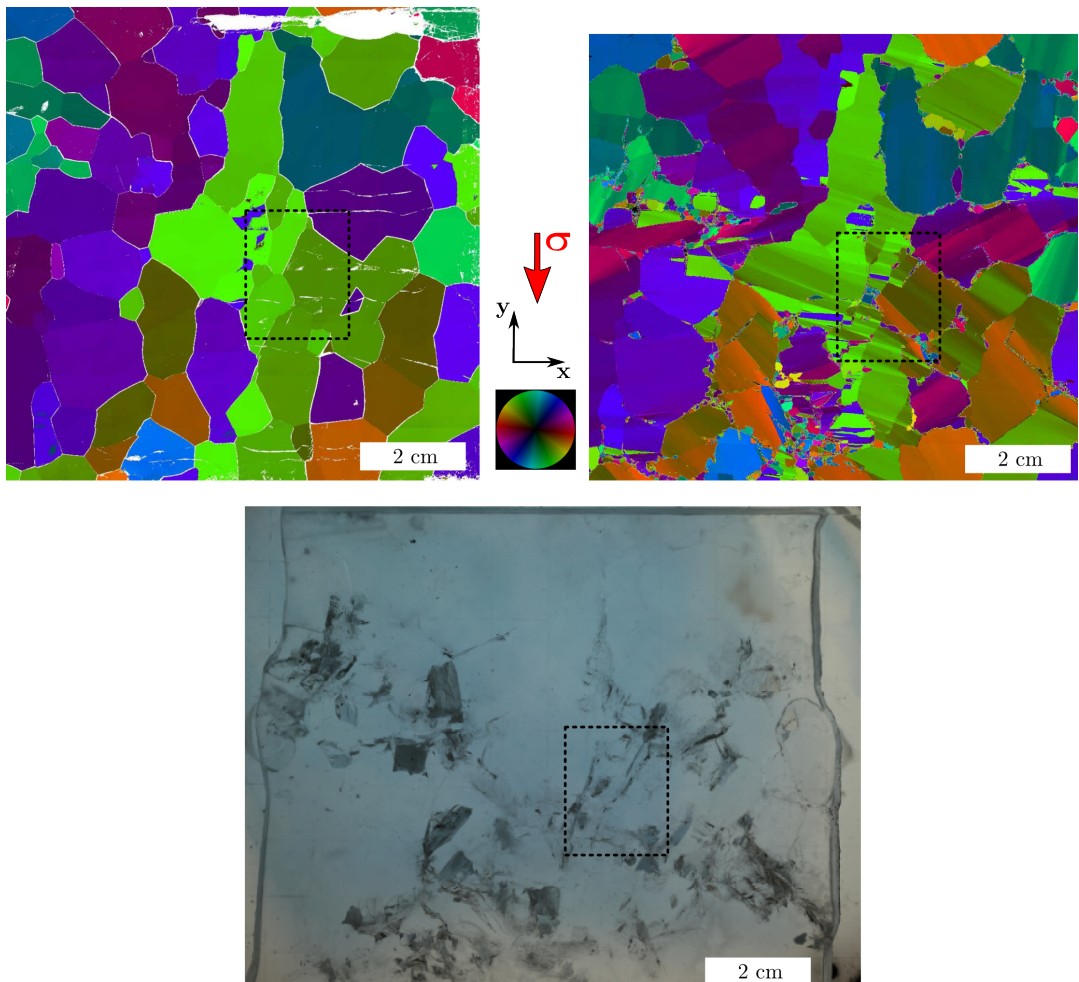

**Figure 3.** *Top:* Microstructure (c-axis orientation color-coded, from AITA analysis) before deformation and after -5.5 % of compressive creep at -7°C under 1 MPa. *Right:* Raw picture of the sample taken in natural light at the end of the compressive test. Black areas result from light diffusion by cracks and de-cohesion features. The dashed black rectangle surrounds the area studied in details in the paper.

if the AITA observation based on thin sections (about 0.3 mm thick). A lot of subgrain boundaries similar to the tilt and kink bands characterized in (Chauve et al., 2015, 2017) are visible after deformation. A tilt band is composed of basal edge dislocations and can accommodate a large misorientation, as observed here. A kink band is composed of two nearby tilt bands that accommodate opposite misorientations. Kink bands were shown by Chauve et al. (2015) to coincide with local shear strain.

5  Intra-granular cracks (cracks 1, 3 and 4) and cracks along grain boundaries (crack 2) are observed. Observed intra-granular cracks do not always cross the entire grain, such as for crack 1 that seems interrupted in the middle of the grain (figure 5). Such a final microstructure evidences strong strain heterogeneities resulting from stress concentrations at grain boundaries and within grain interiors.





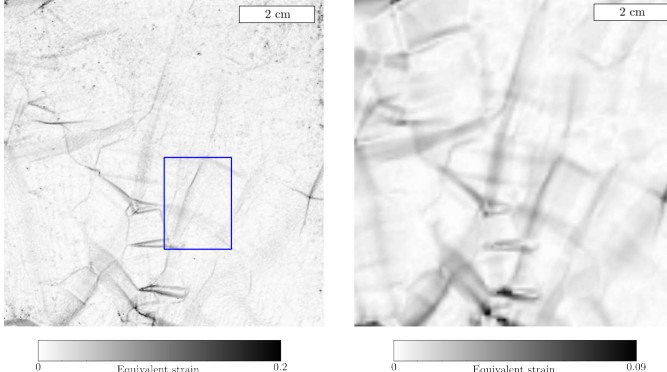

**Figure 4.** Map of equivalent strain ($\varepsilon_{eq}$) after $-0.47\%$ of compressive deformation. The blue rectangle surrounds the area studied in details in the paper. *Left:* Spatial resolution of $0.19\ mm.pix^{-1}$, *Right :* Spatial resolution of $0.76\ mm.pix^{-1}$.

In the following, we track the history of formation of the 4 cracks labelled in figure 3 by analyzing the strain field evolution

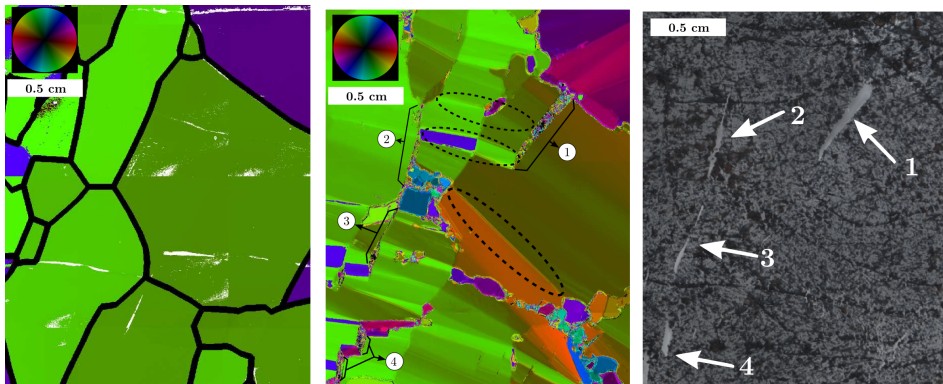

**Figure 5.** Studied area from the sample of figure 3. *Left:* Initial microstructure measured by AITA. The black lines show the superimposed grain boundaries. The irregular white lines and small white areas at are measurements artifacts. *Center:* Microstructure after deformation measured by AITA. Crack locations are highlighted by black full lines and labeled. Areas where recrystallization by sub-grain rotation took place are surrounded by black dashed ellipses. *Right:* Raw picture of the surface speckle at the end of the test where cracks 1 to 4 can be seen by speckle discontinuities.

through the principal strain components, such as in (Chauve et al., 2015).

Within this area of interest, the deformation before the apparition of any crack is localised in two main bands, one crossing the full area at about 10 to 20° from the vertical (compression) direction, and another one in the bottom part of the area, nearly perpendicular to the first one (figure 6, *left*). Both bands are following some boundaries, and crossing some grains. The principal strain components are typical of a local pure shear configuration.

The accumulated strain field just before (at $\varepsilon_{yy}$ = -1.35%), and just after the opening of cracks 1, 2 and 3 (at $\varepsilon_{yy}$ = -1.46%)





is shown in figure 6. As an illustration, the location of cracks 2 and 3 is surrounded by a blue ellipse, and the small dark dots within this ellipse attest of an apparent high equivalent strain due the speckle modifications from cracking. Cracks 2 and 3 occurred at the side of the main deformation bands, but not on these bands. Crack 2 appears to be the closest to the grain boundary, although grain boundaries cannot be positioned precisely enough on top of the image after deformation.

5    As mentioned in part 2, maps of accumulated strain include some noise, and likely some out-of-plane component of strain. In order to provide a more precise description of the relation between strain field and crack opening, the following analyses will be performed based on strain field increments measured during adapted macro strain increments.

As a departure point, strain field during increment of macro strain between -1.34% and -1.35%, just before any visible crack

10    event, is shown in figure 7, by both $\varepsilon_{eq}$ (grey colorscale) and the projections of the principal strain components (arrows). During this strain increment, strain field is very similar to the one accumulated during the entire experiment before cracking events (figure 6, *left*). Blue ellipses were drawn on the position of future opening of cracks 1, 2 and 3. Local strain within the ellipses is low compared to the one accommodated by the two main bands, and we observe no precursor to the cracks nucleation from the strain field pattern (within the limit of the available resolution).

Strain field measured during increments at later steps of the experiment is presented on the different parts of figure 8. Cracks

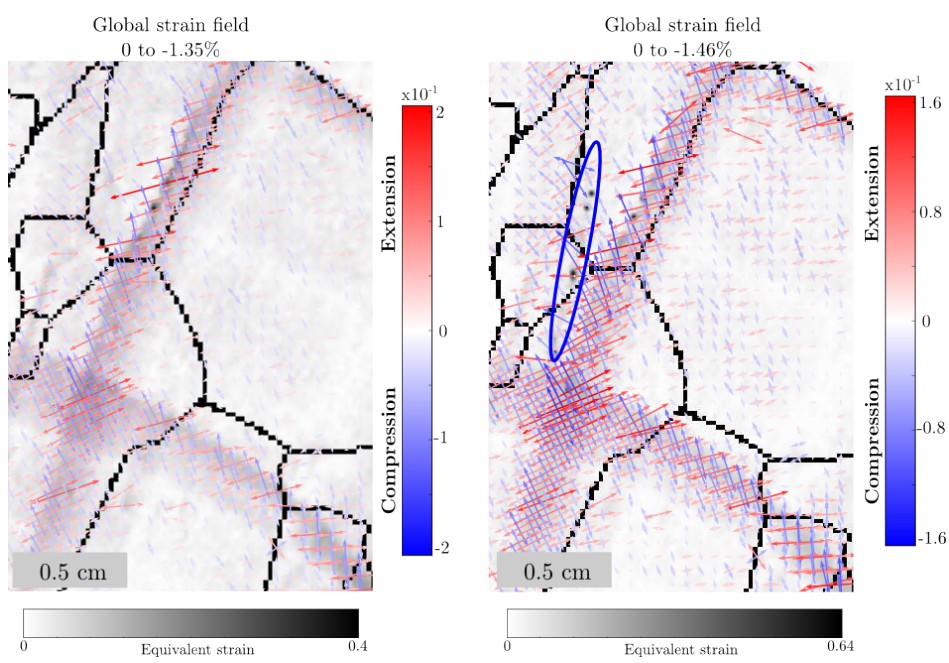

**Figure 6.** Evaluation of the total accumulated strain field in the studied area of the sample of figure 3 represented by $\varepsilon_{eq}$ and by the principal components. *Left:* Total strain accumulated after -1.35% of macro strain. *Right:* Total strain accumulated after -1.46% of macro strain, just after the crack formation. The blue ellipse surrounds the area of formation of crack 2.

1, 2 and 3 are first observed to open between $-1.35\%$ and $-1.46\%$ of macro strain (figure 8a). Cracks can be visualised on




the speckle as discontinuities, and the DIC calculation provides an apparent strain characterised by a pure extension which provides the main direction of the crack opening. Crack 1 to 3 must have been opened mainly in mode I since no shear component was measured by DIC in the area of crack formation before the opening (figure 7), again in the limit of resolution of our observations. During this strain increment, associated with the crack opening, the strain is still localized in the nearly

horizontal band. The nearly vertical deformation band, which was accommodating a lot of deformation before the cracks start to open is not active anymore (figure 8).

During the next increment (figure 8b), cracks continue opening, and a clearer tension component is evaluated by DIC around the crack sides. From the final microstructure picture (figure 6), we see that the top right part of crack 1 is connected to a grain boundary but the bottom part of this crack remains inside the grain, while no clear strain localisation can be observed at this

position (within our limit of accuracy).

About 30 min later, between macro strain of $-2.40\%$ and $-2.59\%$ (figure 8c), the strain field evaluation (together with speckle observation) tends to show that cracks 1 and 2 are still expending when crack 3 remains stable, since DIC calculation shows no more tensile strain components on the area of crack 3.

During this increment crack 4 is appearing at the bottom left, and two new deformation bands appear at the lower tip of crack

2, and at the bottom tip of crack 3, both being parallel to the main transverse deformation band observed from the beginning of this sequence. These new lines of strain localisation end-up joining each others and the initial transverse deformation band between macro strain of $-3.12\%$ and $-3.37\%$ (figure 8(d)).

By looking at the final microstructure (figure 5), these new deformation bands appear to be localised in an area where new grains recrystallized. The strain localized around the new boundaries formed by nucleation, and this strain redistribution seems

to be responsible for the closure of crack 3 visible on the speckle image, also evidenced from the pure compressive strain component in the crack area (blue arrows figure 8d). Such a crack closure very likely comes from a local compressive stress component. By being able to follow strain field evolution during the test, we can well estimate that the new grains have formed after the apparition of the cracks.

During the last increment of deformation (between $-5.05\%$ and $-5.50\%$ of macro strain), pure compressive principal strain

components are calculated in most of the observed crack discontinuities (figure 9). Together with the visual observation of crack evolution on the speckle images, this observation tend to show a crack closure mechanism. During this last increment, the strain field is also characterized by several new lines of strain localisation in the area, which, since they are characterized by pure shear principal strain components, are related to deformation bands (figure 9). By observing the final microstructure, we can attribute these strain localisations to the formation of high angle sub-grain boundaries and kink bands. There likely

locations are surrounded by dashed black ellipses in figures 5 and 9 to facilitate the observation. In particular, the two kink bands marked by the top black dashed ellipses seem to be localized at the tips of cracks 2 and 1. Please note that crack 1 bottom tip localized in the grain interior strongly coincides with the edge of a high angle subgrain boundary.

To summarize, by measuring the strain field evolution during the onset of tertiary creep, at the ductile to brittle transition, we were able to follow crack formation close to grain boundaries and within grain interiors, and there consequences on the local

strain field. Some cracks appear at the side of high strain localisation bands, where stress must have concentrated in a "hard"





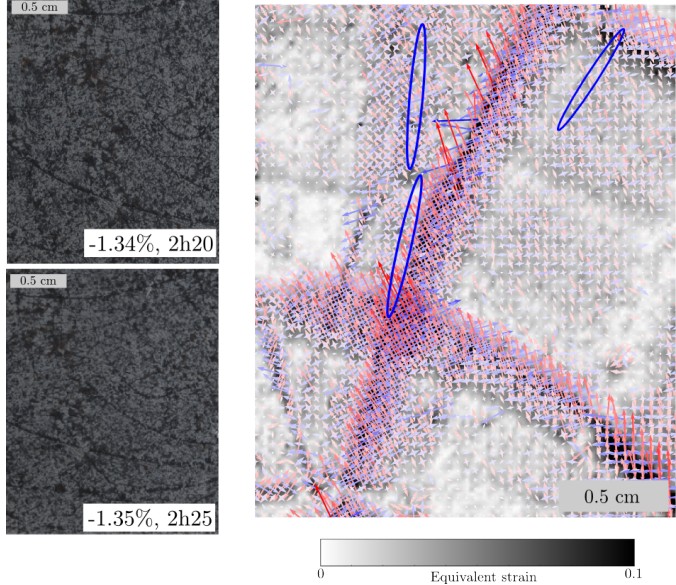

**Figure 7.** Strain field increment during the 5 mn before the apparition of cracks between $-1.34\%$ and $-1.35\%$ of macroscopic strain. *Left:* Pictures of the speckled surface used for the DIC. *Right:* Principal component of the strain field superimposed on the equivalent strain field ($\varepsilon_{eq}$).

zones for deformation. Following crack opening, we observe a strong redistribution of the local strain, with the disappearance of one of the major localisation band. Besides, we show that stress concentration at crack tips can be efficiently relaxed by dynamic recrystallization mechanisms (nucleation and subgrain boundary formation), and that the stress redistribution induced by these mechanisms can lead to the closure of cracks during the test. The occurrence of dynamic recrystallization mechanisms

is, here, strongly enhanced by the high homologous temperature conditions of the experiment.

## 4   Discussion - Mechanisms to relax local stress concentration

During compressive tests on an isotropic material, the maximum shear stress occurs at 45° from the compression direction (Tresca criterium). The boundary conditions therefore induce, by themselves, some stress concentration. For an anisotropic configuration, the polycrystalline structure associated with a strong viscoplastic anisotropy of the single crystal will induce a

redistribution of stress, depending on the orientation relation between grains. Such a redistribution has been simulated by full field crystal plasticity approaches by Lebensohn et al. (2004) and Grennerat et al. (2012) for instance. Although stress field is not experimentally accessible so far, these modeling results were validated by a comparison between predicted and measured strain field magnitudes and heterogeneities (Grennerat et al., 2012).

At the onset of tertiary creep in laboratory deformed ice, strain-rate increases thanks to accommodating processes. As sum-

marized by Schulson and Duval (2009), depending on the deformation conditions (temperature, imposed stress or imposed




**Figure 8.** Four steps of 5 mn strain field increment during crack opening. *Left:* Pictures of the speckled surface used for the DIC. *Right:* Principal component of the strain field superimposed on the equivalent strain field. *(a)* Increment between −1.35% and −1.46%. *(b)* Increment between −1.46% and −1.60%. *(c)* Increment between −2.40% and −2.59%. *(d)* Increment between −3.12% and −3.37%.

strain-rate), accommodation can take place through dynamic recrystallization or micro-cracking.

There exists, to our knowledge, no direct observations of the effect of these mechanisms on the redistribution of strain and therefore on local stress relaxation. The results presented here fill this gap by exploring the ductile to brittle transition where micro-cracking and plasticity can coexist. A common feature with previous observations made by Grennerat et al. (2012) and
5 Chauve et al. (2015), is the strong strain heterogeneities, with local strains reaching more that 10 to 20 times the macroscopic strain. Although influenced by the boundary conditions, grain interactions tend to deviate the strain concentration from the





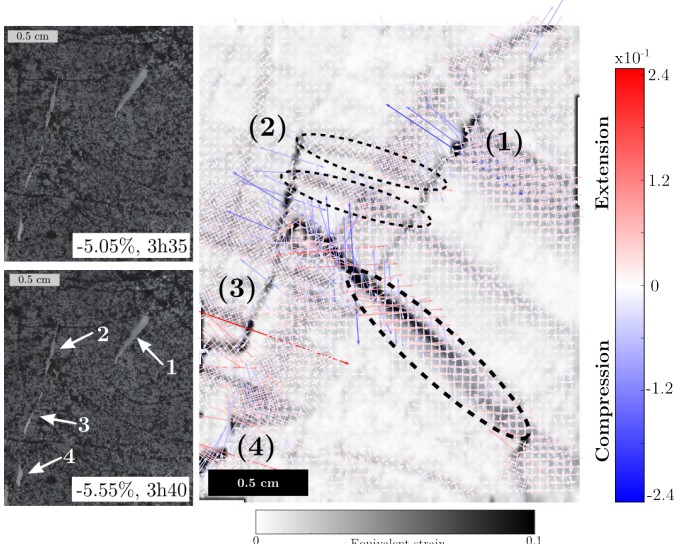

**Figure 9.** Increment of deformation during the last 5 mn of the test (between $-5.05\%$ and $-5.55\%$). Kink band formation (dashed black ellipses surround areas) at crack tips and crack closure are observed. *Left:* Pictures of the speckled surface used for the DIC. *Right:* Principal component of the strain field superimposed on the equivalent strain field.

main $45°$ directions.

Conditions imposed during the experiment presented here induced local cracking at the onset of tertiary creep (which occur before 1% of macro strain for the sample studied very likely because of the influence of a non isotropic texture and of a columnar microstructure). Most of the local cracks observed were intragranular. Cracks appeared in areas nearby strain localization

bands, but not within these bands, as evidenced by figure 6 and by comparing figures 7 and 8(a) (cracks 1 and 2). These observations highlight the fact that local stresses are concentrated at the side of high strained region, because of strain incompatibilities between regions of different orientations, and most likely because of locally low Schmid factors. The likely impact of low local Schmid factors might be strengthened by the strong viscoplastic anisotropy of ice that renders some orientations strongly unfavourable for basal dislocation slip.

Crack formation is relaxing the local stresses, but meanwhile, stress concentration is translated at the crack tips. Previous studies on columnar ice performed at higher strain-rate ($\dot{\varepsilon} = 4 \times 10^{-3} \ s^{-1}$) but similar temperature ($T = -10°C$) (Batto and Schulson, 1993; Iliescu and Schulson, 2004) evidenced the typical mechanism of wing-crack formation at crack tips. Wing cracks appear as the result of tensile stress concentration at the crack tips and can lead to the overall failure of the sample by propagating through it, or by connecting to other cracks. Recently, a similar mechanisms of wing cracks propagation has

been characterized by DIC in a soft rock by Nguyen et al. (2011), and they were able to quantify the different fracture modes (opening, closing and shearing) thanks to local strain measurements.

The experiment presented here being performed in conditions equivalent to a lower strain rate (although through imposed load





conditions) compared to Batto and Schulson (1993), the stress concentration at the crack tips is not relaxed by the formation of wing cracks but by plasticity mechanisms in the creep zone at the tip. Processes by which dislocations can nucleate and propagate at crack tips in a ductile to brittle transition are reviewed in (Argon, 2001). He shows that both nucleation of dislocations at crack tip, and the mobility of the nucleated dislocations come into play to induce the stress relaxation responsible for a

crack arrest. Considering the high temperature conditions of our experiments, the dislocation multiplication leads to dynamic recrystallization mechanisms to occur in the creep zone at the crack tips. Indeed, nucleation of new grains is observed at the bottom crack tip of crack 2 (figure 8 c and d) and dislocation substructures as subgrains are being formed around crack tips of cracks 1 and 2 (figure 9). These observations reveal that plasticity-driven recrystallization mechanisms are efficient to relax the local tensile stresses initiated at the crack tips.

Local stresses associated with grain interactions during deformation of ice was indeed shown to be strongly heterogeneous, and to be responsible for the initiation of subgrain boundaries at the end of primary creep (Piazolo et al., 2015). Observation of crack initiation nearby grain boundaries and within grain interior is another evidence of such a local stress concentration. Besides, our observations enable us to follow the "story" one step further by showing the role of recrystallization mechanisms in the redistribution of local stresses.

By following the strain field evolution all along the tests, we observe the closure of some part of the cracks, in areas where nucleation and subgrain boundary formation were the most active. The crack closure is evidenced by the representation of principal strains which directions evolve from a tension component to a compressive component that ensures the recovering of continuity (figure 8d). In order to obtain a local closure of cracks, the stress field must provide a local compression component, perpendicular to the crack surface. The new microstructure formed by recrystallization mechanisms must therefore drive a

redistribution of the local stress field to enable such a modification, still compatible with the macroscopic stress conditions.
Similar observations of a ductile to brittle transition in Olivine driven by plasticity mechanisms was deeply studied by Druiventak et al. (2011). In samples deformed at 20°C, 300°C and 600°C they observed microcracking at grain boundaries and in the grain interiors, but also arrays of dislocations related to crystal plasticity. Similarly to our observations, at the highest temperature, plasticity took place in the form of strongly misoriented undulatory extinctions (associated with various types of

dislocations), deformation lamellae, and 3D dislocation cells inducing strong modifications of the microstructure. Our results therefore present some interest beyong the ice community. Similar procedures could very interestingly be applied to a wide range of materials in order to estimate the role of the level of plastic anisotropy on strain localization and on the efficiency of plasticity-driven recrystallization mechanisms to relax the local stress field at crack tips.
On top of the mechanical meaning of these observations, we highlight the fact that, since we were able to follow local crack

closure during the test, care must be taken when performing experimental tests in conditions close to the ductile to brittle transition (typically at strain rates above $10^{-6}s^{-1}$, or compressive stress above 0.9 MPa), and at high temperature. Microcracking can influence the local stress relaxation, and therefore the mechanical response, without leaving any track in the final microstructure.





## 5   Concluding remarks

The present work reveals, for the first time in ice, the evolution of the local strain field during the onset of tertiary creep, in conditions where local cracking occurs to relax the stress field. This observation was made possible by taking advantage of samples with 2D-1/2 microstructures from which surface observations follow bulk behavior.

While strain field localises into large bands compared to the grain dimensions, cracks appear nearby but not on the strain localisation zones, where deformation by dislocation glide must have been impeded by low Schmid factor conditions.

Relaxation of the local stress field by crack opening results in a local redistribution of the strain field, as evidenced by the weakening of some deformation bands. At the crack tips, where stress concentrates, dynamic recrystallization mechanisms are observed in the plastic zone (nucleation and high angle grain boundary formation). These mechanisms induce a strong redistribution of the local strain field such as already observed by Chauve et al. (2015).

While induced by local stress concentration at crack tip, recrystallization mechanisms in turn generate a stress field redistribution as a result of the microstructure modification. This redistribution is indirectly evidenced by the modification of the measured strain field in the area, but also by the original observation of local crack closure, that should result from a local compressive stress, in place of the initial tensile stress responsible for the crack opening.

While stress field is not directly evaluated, we show here that coupling strain field measurements to microstructure analyses is a powerful tool to follow stress redistribution during creep tests at the ductile to brittle transition. In particular, micro cracking and dynamic recrystallization mechanisms can coexist, the later being efficient to relax stress concentration at crack tips. One may therefore be careful when working at the frontiers of this transition since recrystallization can hide local cracking on the final microstructures.

*Author contributions.*   T. Chauve, D. Georges and C. Lachaud performed the laboratory experiments. D. Georges and T. Chauve provided the data treatment. M. Montagnat and T. Chauve analysed the data and wrote the paper. P. Vacher provided some support for the DIC analyses and interpretation.

*Competing interests.*   There is no competing interest

*Acknowledgements.*   Financial support by the French "Agence Nationale de la Recherche" is acknowledged (project DREAM, ANR-13-BS09-0001-01). This work benefited from support from institutes INSIS and INSU of CNRS. It has been supported by a grant from Labex OSUG@2020 (ANR10 LABEX56) and from INP-Grenoble and UJF in the frame of a proposal called "Grenoble Innovation Recherche AGIR". Support from the imagery center IRIS of Grenoble-INP is acknowledged.



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
