# Peer review of "Strain heterogeneities at the ductile to brittle transition; a case study on ice."

_Solid Earth, 2017_

## Referee Comment (RC1) · Anonymous Referee #1 · 21 Apr 2017

In the work described in this MS, the authors investigate, using the method of Digital Image Correlation (DIC), the localization of plastic strain and of cracking within one specimen of S2 freshwater ice as it shortened by up to 5.5% under a set of creep conditions just on the ductile side of the ductile-to-brittle transition; specifically, under a compressive stress of 1 MPa at -7 o C (or 0.97 Tm). They present observations which show that strain is concentrated within a few bands, by a factor as high as 10 to 20. In addition—and this is the novelty of the MS–they claim to see evidence of dynamic recrystallization at the tips of short, deformation-induced cracks that formed outside the regions of localized plastic flow. Recrystallization, they state, serves to relax and then to redistribute stresses that develop within the crack-tip plastic zone and thus, presumably, to stabilize cracks against propagation. The work could be seen to support an earlier model of the DB transition (Renshaw and Schulson, 2001), even

though that model does not specify the need for dynamic recrystallization, and thus to add further detail about a phenomenon that is important to the inelastic behavior of a variety of materials. That said, the MS leaves something to be desired.

The major shortcoming is unambiguous evidence of dynamic recrystallization at crack tips. The claim is made (p.10, lines 18-19; p.14, lines 5-8) that cracks-2 & 3 of Fig 8c-8d appear to be localized within an area where new grains recrystallized. While it is quite reasonable to expect that recrystallization could occur within the plastic zone at the tips of cracks, particularly within material as warm as that examined here, the evidence to support this point—the key point of the MS—is not compelling.

Another shortcoming, perhaps more an oddity that a weakness, is the apparent absence of shear deformation within the near-vicinity of cracks. In Figure 8 strain near the three cracks is shown to be predominantly tensile. Yet the cracks are inclined to the direction of loading and so one would have expected a shear stress to act in their plane. In the ideal case of no end-constraint (point 3, below), the ratio of shear stress to normal stress is given by R= tangent theta where theta is the angle between the normal to the plane of the crack and the direction of loading; in the real case of end-contraint, R>tangent theta. For crack-3 in Fig 8, for instance, theta~15 degrees so that R>0.25. This is a rather large ratio, begging the question: why is no shear strain detected in the near-vicinity of the three cracks?

Thus, owing to the points noted in the previous two paragraphs, this MS as presently developed should not be published. However, the authors should be encouraged to pursue their work, for the presentation of unambiguous and compelling evidence of the main point they have in mind, assuming it to be correct, would be a positive contribution to the literature. In so doing, they should consider and then address the following points:

1. In calculating strain from relative displacement of points on a speckle pattern using the DIC method, what precaution was taken to ensure that the only movement detected

from one image to the next was through deformation of the ice? In other words, to what extent did vibration and other extraneous movements of the camera contribute to apparent displacement and hence to inelastic strain?

2. Identify using an arrow "decohesion features" in Fig 3c, and then define them. Are they the kind of feature reported by Picu and Gupta (Acta Mater., 43(10), 3791-3797 (http://dx.doi.org/10.1016/0956-7151(95)90163-90) and by Weiss and Schulson (Phil. Mag. A, 80(2), 279-300 (dx.doi.org/10.1080/01418610008212053).

3. Could the deformation bands shown in Fig.4 and elsewhere be a result of end-constraint imposed on the square-shaped (9 cm x 9cm) specimen by boundary conditions external to the ice ( i.e.,by the loading platens)? Boundary conditions are mentioned in the Discussion (p.11,12), but more within the context of grain boundaries and their influence on local stress state than within the context of end zones. Given the square shape of the specimen, the entire volume of the ice was effectively confined. To know whether deformation bands are an intrinsic feature of ice creep, experiments need to be run using specimens whose length to width ratio is closer to 3 or more.

4. To Figure 6 and elsewhere where blue (compressive strain) and red (tensile strain) arrows signify the two principal strains, add a scale.

5. In the increment of strain from Fig 8b to 8c, crack-3 appears to close. Closure is claimed (p.10, line 22; p.14, lines 15-20) to be caused by a local compressive stress which is related to the formation of new boundaries formed by nucleation. How exactly would recrystallization develop a compressive stress normal to the plane of crack-3?

6. Typos

The images in Fig.4 should be reversed, in that the one on the left is of the lower spatial resolution.

On p. 7, "if the AITA" should read "of the AITA"

On p.10, lines 29 and 34, "there" should be spelled "their".

On p.14, line 26, "beyong" should be "beyond".

---

## Referee Comment (RC2) · Anonymous Referee #2 · 14 May 2017

This manuscript describes original results on the DIC analyses of poly-crystalline ice under creep deformation. As described in the authors' previous paper (Acta Materialia (2015)), application of the DIC method to ice provides a powerful tool to investigate evolution of strain fields during plastic deformation. In the present manuscript, very interesting results on behavior of local strain fields associated with cracking are presented, and the argument addressed are suitable for publication in Journal SE. However, I found some of the authors' explanations difficult to follow. The manuscript should be improved before acceptance for publication, with considering the following points:

(1) (General comment) For convincing argument, focusing on the experimental results found in the present study, distinguish more clearly those from others already presented in the previous papers by the author(s) and other researchers.

[Figure]

(2) Reconsider the description in 'Abstract'. The main purpose of the study must be to clarify 'the evolution of local strain fields around cracking' by the use of the DIC method as described in the top sentence in 'Abstract', but the description on the most important result obtained by the study is not clear. For example, if the argument is concluded by the last sentence 'A strong interaction between cracking and dynamic recrystallization is therefore evidenced', I wonder if it is a new finding. Such a general phenomenon may be already presented elsewhere. Consider carefully what is the most important finding made by the study. In addition, the title suggests 'strain heterogeneity' for the main topic of the paper but no descriptions about it in 'Abstract' and 'Concluding remarks'.

(3) 'Introduction' should be more concise, with focusing on the main topic of the paper.

(4) I found very interesting results are presented in section 3 'Strain field evolution ....'. It should be emphasized more clearly what is found in the present study, and describe it also in 'Concluding remarks'.

(5) (Line 5 to 6 on p.15. In section 5 'Concluding remarks') What does 'large' mean in 'large bands' (large in width, length, or thickness) ? What is the difference between the 'band' in 'strain field localises into large bands' and the 'zone' in 'strain localization zones'? In addition, the description 'cracks appear nearby but not on the strain localization zones, where deformation by dislocation glide must have been impeded by low Schmidt factor conditions' is difficult to understand. If dislocation glide is impeded in the 'strain localisation zones', how does strain localise into the 'zones' ? A clear-cut description is required in 'Concluding remarks'.

(6) (Line 9 to 10 on p.15) The description 'a strong redistribution of the local strain field such as already observed by Chauve et al. (2015)' should be revised to distinguish more clearly the original results obtained by the present study from the results already presented in other paper to avoid readers' misunderstanding in evaluation of this paper.

(7) (Line 11 to 14 on p.15) This paragraph is not easy to follow because the experimental results (facts) and speculative descriptions are not well distinguished. As a

concluding remark, what was found in the present study should be more clearly described.

---

## Author Comment (AC1) · 20 Jun 2017

June 20th,

We are thankful for very interesting and helpful comments provided by both reviewers. We hope that we were able to make the best use of these comments to improve our paper. In black are the reviewer comments, and the authors response appear in between.

A new version of the paper is attached.

Referee #1

In the work described in this MS, the authors investigate, using the method of Digital

[Figure]

Image Correlation (DIC), the localization of plastic strain and of cracking within one specimen of S2 freshwater ice as it shortened by up to 5.5% under a set of creep conditions just on the ductile side of the ductile-to-brittle transition; specifically, under a compressive stress of 1 MPa at -7°C (or 0.97 Tm). They present observations which show that strain is concentrated within a few bands, by a factor as high as 10 to 20. In addition and this is the novelty of the MS–they claim to see evidence of dynamic recrystallization at the tips of short, deformation-induced cracks that formed outside the regions of localized plastic flow. Recrystallization, they state, serves to relax and then to redistribute stresses that develop within the crack-tip plastic zone and thus, presumably, to stabilize cracks against propagation. The work could be seen to support an earlier model of the DB transition (Renshaw and Schulson, 2001), even though that model does not specify the need for dynamic recrystallization, and thus to add further detail about a phenomenon that is important to the inelastic behavior of a variety of materials. That said, the MS leaves something to be desired.

The major shortcoming is unambiguous evidence of dynamic recrystallization at crack tips. The claim is made (p.10, lines 18-19; p.14, lines 5-8) that cracks-2 & 3 of Fig 8c-8d appear to be localized within an area where new grains recrystallized. While it is quite reasonable to expect that recrystallization could occur within the plastic zone at the tips of cracks, particularly within material as warm as that examined here, the evidence to support this point the key point of the MS not compelling.

RESPONSE :

Our observations are, here, only supported by the observations of the microstructure (with a 20 micron resolution) before and after the experiments. The fact that the microstructure modifications observed close to crack tips are associated with dynamic recrystallization (DRX) mechanisms was based on published recrystallization observations like those recently performed in Chauve et al. 2017, where comparison between similar thin section observations and EBSD observations enabled to clearly assess microstructure changes associated with DRX. Of course, our resolution is not accurate enough to pretend that the new grains or subgrain boundaries appeared exactly at crack tips, but they are located very close to them (new purple grains around crack 4, or blue grains close to crack 3 and crack 2 tips, even light green new grain at the bottom of crack 1) Here, the reviewer mentions figure 8, showing the strain field around cracks, while our DRX observations are based on figures 3 and 5 where new subgrain boundaries and new small grains are visible in the crack region. The sentence p.10, lines 18-19 refers to figure 5, and mentions the relation between deformation bands and the new grains : "By looking at the final microstructure (figure 5), these new deformation bands appear to be localised in an area where new grains recrystallized". In sentences p.14, lines 5-8 (now page 13, line 15) we have changed "at crack tip" by "close to crack tip" to be in better agreement with the limits of our accuracy, and to follow the reviewer's comment.

END

Another shortcoming, perhaps more an oddity that a weakness, is the apparent absence of shear deformation within the near-vicinity of cracks. In Figure 8 strain near the three cracks is shown to be predominantly tensile. Yet the cracks are inclined to the direction of loading and so one would have expected a shear stress to act in their plane. In the ideal case of no end-constraint (point 3, below), the ratio of shear stress to normal stress is given by R= tangent theta where theta is the angle between the normal to the plane of the crack and the direction of loading; in the real case of end-contraint, R>tangent theta. For crack-3 in Fig 8, for instance, theta∼15 degrees so that R>0.25. This is a rather large ratio, begging the question: why is no shear strain detected in the near-vicinity of the three cracks?

RESPONSE

We totally agree with the reviewer, and we were also expecting to observe more shear strain localized in the crack-tip areas. Nevertheless, our observations only show local strain configuration, and to relate it to a local stress state is not straightforward. The

main explanation, that we tried to give in the discussion (but we tried to make it even clearer in the new version) could be related to the strongly anisotropic behavior of ice, and the fact that our microstructure has large grains, and is far for being isotropic. Therefore, as shown already by Grennerat et al. (2012) and Piazolo et al. (2015) (refs in the manuscript) the tensorial local stress state (simulated in these works) can be strongly different from the applied on. On top of that, Chauve et al. (2015) have shown the ability of DIC measurements to evidence local shear strain. The fact that we don't observe a strong shear strain here could therefore be associated with an heterogeneous local stress field, differing from the applied one, and leading mainly to a mode 1 opening of the cracks.

END

Thus, owing to the points noted in the previous two paragraphs, this MS as presently developed should not be published. However, the authors should be encouraged to pursue their work, for the presentation of unambiguous and compelling evidence of the main point they have in mind, assuming it to be correct, would be a positive contribution to the literature. In so doing, they should consider and then address the following points:

1. In calculating strain from relative displacement of points on a speckle pattern using the DIC method, what precaution was taken to ensure that the only movement detected from one image to the next was through deformation of the ice? In other words, to what extent did vibration and other extraneous movements of the camera contribute to apparent displacement and hence to inelastic strain?

RESPONSE

The DIC method used to extract displacement fields, and then strain fields, from image correlations is now very well documented (see Sutton et al. 2009 Image Correlation for Shape, Motion and Deformation Measurements, for instance, but also Vacher et al. 1999 for the 7D software used here). In particular, it was shown that the method

enables to remove any effect of a displacement of the relative position of the camera compared to the sample (in the sample plane). Accuracy tests were performed in the specific case of ice, and with the specific equipment and experiment configuration used here by Grennerat et al. (2012), therefore we refer to this work. And at the end, we can exclude vibrations (negligible in our conditions, creep test, no motor) to provide anything else than noise on top of the signal. Their effect is, therefore, included in the resolution evaluation.

END

2. Identify using an arrow "decohesion features" in Fig 3c, and then define them. Are they the kind of feature reported by Picu and Gupta (Acta Mater., 43(10), 3791-3797 (http://dx.doi.org/10.1016/0956-7151(95)90163-90) and by Weiss and Schulson (Phil. Mag. A, 80(2), 279-300 (dx.doi.org/10.1080/01418610008212053).

RESPONSE

You are right that, in Weiss and Schulson paper, a clear distinction is made between decohesion and cracks. We took this work for granted, assuming that both can exist during brittle deformation of ice (or at brittle to ductile transition like here), but we did not perform the work to distinguish them, since we only focused on what happened around cracks. We could remove the term "decohesion" for clarity? But maybe the best is to add the reference to Weiss and Schulson 2000 paper. See changes p.5 line 4 in the new version of the paper.

END

3. Could the deformation bands shown in Fig.4 and elsewhere be a result of end-constraint imposed on the square-shaped (9 cm x 9cm) specimen by boundary conditions external to the ice ( i.e.,by the loading platens)? Boundary conditions are mentioned in the Discussion (p.11,12), but more within the context of grain boundaries and their influence on local stress state than within the context of end zones. Given the

square shape of the specimen, the entire volume of the ice was effectively confined. To know whether deformation bands are an intrinsic feature of ice creep, experiments need to be run using specimens whose length to width ratio is closer to 3 or more.

RESPONSE

Deformation bands oriented at close to 45° from the compression axis are expected and observed (Grennerat et al. 2012) for this type of experimental setup. Indeed, when the sample is homogeneous (small grain size, RVE), maximum shear stress occurs within these two orientations. Grennerat et al. 2012 have shown that due to the large grain size, to the plastic anisotropy of ice, and to the "non-RVE" configuration of experiments performed on these square-shaped specimen, the bands are deviated from their theoretical orientation. The purpose of this work is clearly not to focus on the intrinsic feature of these deformation bands, and the analysis of their formation in the specific case of ice (since it has already been done by Grennerat et al. 2012 in ice, but also by Heripre et al. 2007, International Journal of Plasticity in metallic materials for instance), but we aim at showing the link between the strain localization, crack mechanisms, and recrystallization. Since it appear that we may not have been cleared enough on this point, we tried to clarify the way we mention boundary conditions in the Discussion.

END

4. To Figure 6 and elsewhere where blue (compressive strain) and red (tensile strain) arrows signify the two principal strains, add a scale.

RESPONSE

Thanks, this has been done.

END

5. In the increment of strain from Fig 8b to 8c, crack-3 appears to close. Closure is claimed (p.10, line 22; p.14, lines 15-20) to be caused by a local compressive stress

which is related to the formation of new boundaries formed by nucleation. How exactly would recrystallization develop a compressive stress normal to the plane of crack-3?

RESPONSE

Local compressive stress is needed to explain the crack closure as written p.14, lines 15-20 "In order to obtain a local closure of cracks, the stress field must provide a local compression component, perpendicular to the crack surface." Hence, to explain such a change in the local stress field from tension (leading to crack opening in mode one) to compression (leading to cracks closing), we assumed that some changes in the local stress field has occurred during the test. The most likely reason for this change in stress field configuration was attributed to the modification of the microstructure, and this change is due to dynamic recrystallization and cracking. Therefore we do not attribute directly the stress changes to recrystallization, but to the modification of microstructure induced by recrystallization, see p.10 lines 27-28 and p.14 lines 8-9. Such redistribution of the local stress field due to microstructure evolution is coherent with previous results of Chauve et al. 2015 (Acta Mater), and is enhanced by the high visco-plastic anisotropy ice.

END

6. Typos The images in Fig.4 should be reversed, in that the one on the left is of the lower spatial resolution.

Ok

On p. 7, "if the AITA" should read "of the AITA"

Corrected

On p.10, lines 29 and 34, "there" should be spelled "their".

Corrected

On p.14, line 26, "beyong" should be "beyond".

Corrected

Please also note the supplement to this comment:
http://www.solid-earth-discuss.net/se-2017-24/se-2017-24-AC1-supplement.pdf

―――――――――――――――――――――

**Supplement:**

**Strain field evolution at the ductile to brittle transition; a case study on ice.**

Chauve Thomas[1], Montagnat Maurine[1], Lachaud Cedric[1], Georges David[1], and Vacher Pierre[2]

[1]Université Grenoble Alpes, CNRS, IRD, G-INP, IGE, F-38041 Grenoble, France
[2]Laboratoire SYMME, Université de Savoie Mont Blanc, BP 80439, 74944 Annecy le Vieux Cedex, France

*Correspondence to:* maurine.montagnat@univ-grenoble-alpes.fr

**Abstract.** This paper presents, for the first time, the evolution of the local heterogeneous strain field around intragranular cracking in polycrystalline ice, at the onset of tertiary creep. Owing to the high homologous temperature conditions and relatively low compressive stress applied, stress concentration at crack tips is relaxed by plastic mechanisms associated with dynamic recrystallization. Strain field evolution followed by Digital Image Correlation directly shows the redistribution of
5   strain during crack opening, but also driven by crack tip plasticity mechanisms and recrystallization. Associated local changes in microstructure induce a modifications of the local stress field evidenced by crack closure during deformation. At ductile to brittle transition in ice, micro cracking and dynamic recrystallization mechanisms can co-exist and interact, the later being efficient to relax stress concentration at crack tips.

**1 Introduction**

10   The evaluation and the characterization of strain heterogeneities is of primary importance in material sciences at various scales of observation. Plastic strain localization in metals play a crucial role on the propagation of fracture and on the response to fatigue conditions, and Portevin-Le-Chatelier is a strong example of plastic strain heterogeneities development during mechanical tests in some metal alloys (see Antolovich and Armstrong (2014) for a review). Similarly, strain heterogeneities and localization are known to strongly influence the rheological behavior of the Earth lithosphere in particular to explain post-
15   seismic deformation (Tommasi et al., 2009; Vauchez et al., 2012).

In the context of ice sheet flow, successive layers of ice with slightly different viscosity can experience different strain history as a result of strain localization initiated by bedrock topography (Paterson, 1994; Durand et al., 2004, 2007). Strain localization can induce flow disturbances that can mix the climatic signal and counteract the search for the oldest ice (Dahl-Jensen et al., 2013; Fischer et al., 2013). These flow disturbances can form as folding, that is observed at large scale from ice-penetrating
20   radar surveys now able to highlight deep stratigraphy (MacGregor et al., 2015; Panton and Karlsson, 2015; Bons et al., 2016), but also at smaller scales from microstructure observations (Jansen et al., 2016).

During ductile deformation of ice in natural or laboratory conditions (at high homologous temperature $\sim 0.97$ $T_m$, low strain rate $\sim 10^{-7} s^{-1}$ and low stress, 0.5 - 1 MPa), plastic deformation is mainly accommodated by the glide of basal dislocations (Duval et al., 1983). The resulting strongly anisotropic viscoplastic behavior of the single crystal (Duval et al., 1983) leads to

the development of strong strain heterogeneities during deformation of polycrystalline ice.

Strain heterogeneities evaluated during transient creep of ice were shown to reach local values higher than 10 times the macroscopic strain, and to settle into bands which dimensions are higher than the grain size. Strain localization bands may follow grain boundaries, but also cross entire grains, and there is no statistical link between the crystallographic orientation and the amount of local strain (Grennerat et al., 2012). These first measurements of strain localization during laboratory experiments were restricted to transient (or primary) creep conditions, in ductile conditions ($\sigma < 0.5$ MPa and T $> 0.97$ T$_m$) and prior to any microstructure modification due to dynamic recrystallization.

More generally, creep of isotropic polycrystalline ice is characterized by a three-stages behavior, with a strong decrease of strain-rate during primary creep, down to a minimum reached at about 1% strain, also called secondary creep, immediately followed by a increase in strain-rate to reach tertiary creep at about 10% strain (see Jacka and Maccagnan (1984); Duval et al. (1983) for instance).

At the onset of tertiary creep, for experiment performed at low strain rate ($< 10^{-7}\ s^{-1}$) or low stress ($< 0.5\ MPa$), dynamic recrystallization mechanisms occur increasingly to relax the kinematic hardening and enable for further ductile deformation to occur (Duval et al., 1983). Dynamic recrystallization leads to strong modification in microstructure and texture (Duval, 1979; Jacka and Maccagnan, 1984; Montagnat et al., 2015) through various mechanisms such as nucleation of new grain, polygonisation associated with subgrain boundaries and bulging, as characterized by cryo-EBSD (Chauve et al., 2017). While Piazolo et al. (2015) showed that sub-grain boundary formation such as kink bands could be correlated with heterogeneities of local stress (simulated with a full-field crystal plasticity code, CraFT), Chauve et al. (2015) were able to directly associate nucleation mechanisms (polygonisation, bulging) to local modification of the strain field estimated in-situ from DIC measurements.

For creep experiment performed at higher imposed stress (typically above 0.9 MPa), the increase in strain-rate after secondary creep can also be associated with the occurrence of microcracking without a total collapse of the sample (Schulson et al., 1984; Batto and Schulson, 1993; Schulson and Duval, 2009). The local stress field is therefore relaxed by cracks opening at or close to grain boundaries, and depending on the boundary conditions, crack propagation can occur at various rate. This mechanical response is typical of a ductile to brittle transition (Schulson and Buck, 1995; Schulson and Duval, 2009).

In this domain, most of the studies performed so far, some of which mentioned here, focused on macroscopic parameters (deformation and creep curves, evaluation of the effect of temperature and grain size on the strength) and optical observations of the full sample to characterize the nature of the cracks (Batto and Schulson, 1993; Iliescu and Schulson, 2004). From these observations, a theoretical framework was elaborated based on the assumption of the formation of wing cracks at the tip of initial cracks to relax the local stress field (Renshaw and Schulson, 2001). In particular, the conditions required to form these secondary cracks were shown to control the ductile to brittle transition under compression. More recently, Snyder et al. (2016) showed that this model was able to take into account the effect of a prestrain, including recrystallization mechanisms, on the increase of ductile-to-brittle transition strain-rate for ice.

At the ductile to brittle transition, mixture of creep by dislocations and cracking will occur, and it is related to the ability of the material to relax the stress accumulated at the tip of the initial cracks. For instance, Batto and Schulson (1993) showed that a small amount of creep relaxation at the crack tip could be enough to postponed the transition to brittle behavior (in

time or in strain-rate level). The mechanism of relaxation of the stress produced by a crack opening in mode I, through rapid multiplication of dislocations at crack tip was pioneered by Rice and Thomson (1974) and has been reviewed by Argon (2001) for metallic materials. More recently, Martínez-Pañeda and Niordson (2016) were able to simulate the complexity of the effect of strain gradient plasticity on the level of stress at crack tip and on crack-tip blunting. Crack-tip-initiated plasticity is a crucial mechanism to explain a ductile-like behavior at the ductile to brittle transition.

In the present work we use Digital Image Correlation (DIC) technique, already well approved on ice, to evaluate the strain field evolution during a creep experiment on ice polycrystal performed at the ductile to brittle transition. After a brief presentation of the experimental set-up (Part 2), Part 3 will explore stress conditions during which strain-rate increase with tertiary creep results from local cracking. We will see that plasticity is strongly active at crack tips as evidenced by the occurrence of dynamic recrystallization mechanisms. These mechanisms, by modifying the microstructure, indeed play a crucial role to reduce and redistribute the local stress concentration that appears at the crack tips during the ductile to brittle transition.

**2   Experimental set-up**

Unconfined uniaxial creep tests have been carried out on polycrystalline columnar ice samples of type $S2$ (Ple and Meyssonnier, 1997). Parallelepipedic samples ($\sim 90 \times 90 \times 15$ mm$^3$) were built and the column axes were positioned perpendicularly to the larger surface, and to the compression axis (figure 1). By doing so, the samples provide a "2D-1/2" microstructure, from which surface characterization can approximate volume behavior. Sample microstructure and texture were measured using an Automatic Ice Texture Analyser (AITA) (Wilson et al., 2003; Peternell et al., 2011), which is an optical technique measuring the **c**-axis (or optical axis) orientation (azimuth $\theta$ and colatitude $\phi$) with a spatial resolution from 50 to 5 $\mu$m, and an angular resolution of about 3°. Although large areas can be analysed (up to 120×120 mm$^2$), this technique requires the preparation of thin sections of ice ($\sim 0.3$ mm thick), and is then destructive. By taking advantage of the columnar microstructure, we were able to compare *pre-* and *post-* deformation microstructures by carefully extracting thin layers of ice before and after the test (figure 1). Details of the procedure for sample preparation can be found in (Grennerat et al., 2012; Chauve et al., 2015).

During the experiment, DIC analyses were performed over the full surface of the samples by following the procedure adapted to ice by Grennerat et al. (2012). DIC provides in-situ measurements of the displacement and therefore strain field on the sample surface, from the correlation of surface images of a grey-level speckle that follows the sample deformation. By taking advantage of the 2D-1/2 configuration, we assumed the surface strain field to be as representative as possible of the volume deformation. This configuration makes it possible to compare the measured microstructures by AITA (before and after the test) to the strain field evaluated by DIC (figure 1).

The spatial resolution strongly depends on the quality of the speckle, the illumination and the sensitivity of the camera used. In the following experiments, we used a Phase One 80 Mpx camera, the speckle was made of shoe polish that offers a good cohesion with the ice surface, and a good illumination was obtained thanks to two neon lamps. From that, we ended up with

a spatial resolution of 0.19 mm.pix$^{-1}$, and a strain resolution between $3.10^{-3}$ and $4.10^{-3}$ for the different strain components (see table 2). Displacement and total strain data were extracted using the $7D$ software from Vacher et al. (1999). This DIC

| Camera | DIC spatial resolution | DIC strain resolution | | |
|---|---|---|---|---|
| | | $\sigma_{\varepsilon_{xx}}$ | $\sigma_{\varepsilon_{yy}}$ | $\sigma_{\varepsilon_{xy}}$ |
| Phase One 80 Mpx | $0.19\ mm.pix^{-1}$ | $4.10^{-3}$ | $3.10^{-3}$ | $4.10^{-3}$ |

**Table 1.** Characteristics of the DIC measurements.

method provides a set of displacement vectors over a given grid, defined for the DIC calculation as a function of the speckle and picture qualities (Vacher et al., 1999). From the displacement field components, the total strain components are extracted

5   by using Green-Lagrange expression. Please note that the elastic and plastic components can not be separated, and strain field refers to the total strain field. In the case of ice, elasticity is very low and nearly isotropic, and can be neglected (Schulson and Duval, 2009). In-plane components of strain are therefore provided ($\varepsilon_{xx}$, $\varepsilon_{yy}$, and $\varepsilon_{xy}$), from which an equivalent strain ($\varepsilon_{eq} = \sqrt{\frac{2}{3}\left(\varepsilon_{xx}^2 + \varepsilon_{yy}^2 + 2\varepsilon_{xy}^2\right)}$) and principal strain components are calculated. The later will be plotted along their principal directions in the following figures.

10   Discontinuities such as cracks produce displacements which translation in terms of strain is not direct but could be estimated as shown by Nguyen et al. (2011). In the present study, we simply use the direction of the principal strain components calculated around a crack to interpret the direction of the crack opening (or closing), since the displacement produced is small enough to be followed by the speckle on each side of the crack.

Since all surfaces except the loaded ones remained free (unconfined tests), a slight amount of out-of-plane shear cannot be

15   excluded. The effect of a deformation going out of the plane $xOy$ was estimated in previous analyses performed by Grennerat et al. (2012) and shown to remain low, in the limit of the small macroscopic deformations reached in the present study (less than $5.5\%$). In order to reduce the noise and this out of plane strain effect on the evaluation of the strain evolution during the experiment, we calculated the strain field during short increments of macroscopic deformation of $0.1\%$ to $0.5\%$. Besides, observation of the incremental strain field enables to individualise consecutive events that would be hidden in a strain field

20   calculation integrating the whole experiment duration.

Table 2 summarizes the experimental conditions of the test used as an illustration in this paper, and figure 2 provides the creep curves. The minimum strain rate is reached at about 0.5% of compressive macro strain, slightly before the classical 1% value. This can be attributed to a microstructure effect since our 2D-1/2 samples contain only few grains and do not form a good Representative Volume Elements. In the following, a negative sign will be given to the compressive strain, at the macroscopic

25   and local scale.

**3   Strain field evolution at the ductile to brittle transition**

The macroscopic strain curve reveals an increase in strain rate after -0.5% of $\varepsilon_{yy}$ (vertical) macro strain (figure 2). At -0.5% of macro strain the minimum strain-rate is $5.0 \times 10^{-6}\ s^{-1}$ and at the end of the experiment the strain-rate reaches $8.1 \times 10^{-5}\ s^{-1}$,

[Figure]

**Figure 1.** Scheme of the experimental set-up showing the shape of the sample with the direction of imposed stress (red arrow). The position 0 corresponds to the sample surface (during the test) on top of which the speckle is marked. The microstructure analyzed by AITA prior to deformation is located at about $-0.5\ mm$ and the one after deformation is at about $0.5\ mm$ from the sample surface ($0.5\ mm$ corresponds to the ice thickness needed to make the thin section). The strain-field image, measured at position 0, is added in the front plan for illustration

| Stress (MPa) | Temperature (°C) | Strain rate $(s^{-1})$ | |
|---|---|---|---|
| | | mini $(\varepsilon_{yy} = -0.5\%)$ | end $(\varepsilon_{yy} - 5.5\%)$ |
| 1.0 | $-7$ | $5.0 \times 10^{-6}$ | $8.1 \times 10^{-5}$ |

**Table 2.** Experimental conditions at the ductile to brittle transition for the illustrative test. The minimum creep rate is reached at about -0.5% strain.

evidencing a strong acceleration at the onset of tertiary creep captured here.

The initial microstructure of the sample, the finale one, and an optical observation of the sample at the end of the test are shown in figure 3. Thanks to the transparency of ice, cracks and de-cohesion features can be observed with natural light. They appear as grey and black areas in figure 3. Both features were clearly distinguished and analyzed by Weiss and Schulson (2000).

5  From the c-axis orientation color-scale, one can see that the initial texture is not isotropic. On top of the expected columnar grain-shape effect, we therefore expect, as observed (figure 2), a macroscopic mechanical response different from the one of an isotropic granular sample.

The global strain field measured prior to any visible crack opening on the speckle, at $-0.5\%$ of macro strain (at the minimum creep rate) is represented in figure 4 via the equivalent strain $\varepsilon_{eq}$ at two different spatial resolutions, in order to illustrate

10  the structure of strain heterogeneities. Similarly to what was already observed by Grennerat et al. (2012), the deformation is organized into bands crossing most of the sample. The main orientation of the bands is about 20 to 30° from the compression

[Figure]

**Figure 2.** Evolution of the macroscopic strain and strain-rate measured by DIC. Values before $10^{-3}$ macro strain were not calculated.

direction. Local equivalent strain amplitude in the deformation bands can reach more than 10%, for a $\varepsilon_{yy}$ macro strain of about -0.5%.

In the following a focus will be given on a small area located within the dashed black rectangle of figure 3. The initial microstructure and orientations of the grains in this area, the final microstructure where cracks, subgrain boundaries, and small nucleated grains appear, and a picture of the speckle from the surface of the sample where crack locations are visible (arrows 1 to 4) are shown in figure 5. Very small grains visible in the cracks are artifacts from the thin sectioning process (shaving produces small ships that fill the crack interior), but new grains from dynamic recrystallization mechanisms (DRX) can be distinguished away from the crack interior. See for instance the new blue area in between bottom of crack 2 and top of crack 3. Also the dark blue grain at the bottom of crack 3, and the pink-purple ones surrounding crack 4. None of these new grains were pre-existing in the initial microstructure, and therefore illustrate DRX nucleation. Grain boundaries surrounding new small grains also appear perturbed because of intrinsic limitation of the AITA observation based on thin sections (about 0.3 mm thick). A lot of subgrain boundaries similar to the tilt and kink bands characterized in (Chauve et al., 2015, 2017) are visible after deformation. A tilt band is composed of basal edge dislocations and can accommodate a large misorientation, as observed here. A kink band is composed of two nearby tilt bands that accommodate opposite misorientations. For instance, a highly misoriented tilt band is visible as a sharp transition between orange and light brown close to cracks 2 and 3 (ellipse). The color transitions from green to blue to green, and from brown to green to brown between cracks 2 and 1 illustrate kink bands.

New strain localization patterns were shown to occur close to new grain boundaries and kink bands during DRX by Chauve et al. (2015). Their formation can therefore be followed indirectly by in-situ DIC measurements.

Intra-granular cracks (cracks 1, 3 and 4) and cracks along grain boundaries (crack 2) are observed. Observed intra-granular cracks do not always cross the entire grain, such as crack 1 that seems interrupted in the middle of the grain (figure 5). Such a final microstructure evidences strong strain heterogeneities at grain boundaries and within grain interiors.

In the following, we track the history of formation of the four cracks labelled in figure 3 by analyzing the strain field evolution

[Figure]

**Figure 3.** *Top:* Microstructure (c-axis orientation color-coded, from AITA analysis) before deformation (left) and after -5.5 % of compressive creep at -7°C under 1 MPa (right). *Bottom:* Raw picture of the sample taken in natural light at the end of the compressive test. Black areas result from light diffusion by cracks and de-cohesion features. The dashed black rectangle surrounds the area studied in details in the paper.

through the principal strain components, such as in (Chauve et al., 2015). The principal strain component representation enables to specify the compression, tension and shear component of the local strain.

[revised manuscript text omitted]

By looking at the final microstructure (figure 5), these new deformation bands appear to be localized in an area where new 10   grains recrystallized. Since we follow the strain field evolution during the test, we are able to verify that the new grains have formed after the apparition of the cracks as in(Chauve et al., 2015).

In the same time, crack 3 is closing, as evidenced by the thinning of the corresponding white zone in the speckle image. Strain in the crack 3 area turns into a pure compressive component (blue arrows, figure 8d), that is likely to be responsible for this crack closure. Similarly, during the last increment of deformation (between $-5.05\%$ and $-5.50\%$ of macro strain), pure 15   compressive principal strain components are calculated in most of the observed crack discontinuities (figure 9). Together with the visual observation of crack evolution on the speckle images, these observations reveal a crack closure mechanism.

During this last increment, strain field is also characterized by several new bands of strain localization in the area (figure 9). By observing the final microstructure, we can attribute this strain localization to the formation of high angle sub-grain boundaries and kink bands. Their likely locations are surrounded by dashed black ellipses in figures 5 and 9 to facilitate the observation. In 20   particular, the two kink bands marked by the top black dashed ellipses seem to be localized at the tips of cracks 2 and 1. Please note that crack 1 bottom tip localized in the grain interior strongly coincides with the edge of a high angle subgrain boundary.

    To summarize, by measuring the strain field evolution during the onset of tertiary creep, at the ductile to brittle transition, we were able to follow crack formation close to grain boundaries and within grain interiors, and there consequences on the local strain field. Some cracks appear at the side of high strain localization bands, where stress must have concentrated in "hard" 25   zones for deformation. Following crack opening, we observe a strong redistribution of the local strain, with the disappearance of one of the major localization band. Besides, we show that stress concentration at crack tips can be efficiently relaxed by dynamic recrystallization mechanisms (nucleation and subgrain boundary formation), and that the stress redistribution induced by crack opening and microstructure changes due to DRX mechanisms can lead to the closure of cracks during the test. The occurrence of dynamic recrystallization mechanisms is, here, strongly enhanced by the high homologous temperature 30   conditions of the experiment.

**4   Discussion - Mechanisms to relax local stress concentration**

During compressive tests on an isotropic material, the maximum shear stress occurs at $45°$ from the compression direction (Tresca criterium). For material with plastic anisotropy such as ice, a redistribution of stress is expected to occur that depends

[Figure]

**Figure 7.** Strain field increment during the 5 mn before the apparition of cracks between −1.34% and −1.35% of macroscopic strain. *Left:* Pictures of the speckled surface used for the DIC. *Right:* Principal component of the strain field superimposed on the equivalent strain field ($\varepsilon_{eq}$).

on the orientation relation between grains. Such a redistribution has been simulated by full field crystal plasticity approaches by Lebensohn et al. (2004) and Grennerat et al. (2012) for instance. Although stress field is not experimentally accessible so far, these modeling results were validated by a comparison between predicted and measured strain field magnitudes and heterogeneities (Grennerat et al., 2012).

5   At the onset of tertiary creep in laboratory deformed ice, strain-rate increases thanks to accommodating processes. As summarized by Schulson and Duval (2009), depending on the deformation conditions (temperature, imposed stress or imposed strain-rate), accommodation can take place through dynamic recrystallization or micro-cracking.

There exists, to our knowledge, no direct observations of the effect of micro-cracking on the redistribution of strain and therefore on local stress relaxation. The results presented here fill this gap by exploring the ductile to brittle transition where

10   micro-cracking and plasticity can coexist. A common feature with previous observations made by Grennerat et al. (2012) and Chauve et al. (2015), is the strong strain heterogeneities, with local strains reaching more that 10 to 20 times the macroscopic strain. Although influenced by the boundary conditions, grain interactions tend to deviate the strain concentration from the main 45° directions. While Grennerat et al. (2012) work remained in the primary creep regime, and mostly concentrate on sample-scale field characterizations, Chauve et al. (2015) went a step further and associated DRX mechanisms to explain the

15   interplay between local changes in microstructure and strain field evolution. In particular, strain was shown to re-localize close to the newly formed grain boundaries and subgrain boundaries. This has also been observed in the present study.

Compared with previous works, conditions imposed during the experiment presented here induced local cracking at the onset of tertiary creep (which occurs before 1% of macro strain for the sample studied very likely because of the influence of a non isotropic texture and of a columnar microstructure). Most of the local cracks observed were intragranular. Cracks appeared in

[Figure]

**Figure 8.** Four steps of 5 min strain field increment during crack opening. *Left:* Pictures of the speckled surface used for the DIC. *Right:* Principal component of the strain field superimposed on the equivalent strain field. *(a)* Increment between $-1.35\%$ and $-1.46\%$. *(b)* Increment between $-1.46\%$ and $-1.60\%$. *(c)* Increment between $-2.40\%$ and $-2.59\%$. *(d)* Increment between $-3.12\%$ and $-3.37\%$.

areas nearby strain localization bands, but not within these bands, as evidenced by figure 6 and by comparing figures 7 and 8a (cracks 1 and 2). These observations highlight the fact that local stresses can be concentrated at the side of high strained region. This can result from strain incompatibilities between regions of different orientations, with regions with locally low Schmid factors (relative to the local stress tensor) behaving as solid inclusions in composite materials. The likely impact of low local Schmid factors might be strengthened by the strong viscoplastic anisotropy of ice that renders some orientations strongly unfavourable for basal dislocation slip.

[Figure]

**Figure 9.** Increment of deformation during the last 5 min of the test (between $-5.05\%$ and $-5.55\%$). Kink band formation (within dashed black ellipses) at crack tips and crack closure are observed. *Left:* Pictures of the speckled surface used for DIC. *Right:* Principal component of the strain field superimposed on the equivalent strain field.

Crack formation is relaxing these high local stresses, and meanwhile, stress concentration is translated at the crack tips. Previous studies on columnar ice performed at higher strain-rate ($\dot{\varepsilon} = 4 \times 10^{-3}\ s^{-1}$) but similar temperature ($T = -10°C$) (Batto and Schulson, 1993; Iliescu and Schulson, 2004) evidenced the typical mechanism of wing-crack formation at crack tips. Wing cracks appear as the result of tensile stress concentration at the crack tips and can lead to the overall failure of the sample

5   by propagating through it, or by connecting to other cracks. Recently, a similar mechanism of wing cracks propagation has been characterized by DIC in a soft rock by Nguyen et al. (2011), and they were able to quantify the different fracture modes (opening, closing and shearing) thanks to local strain measurements.

The experiment presented here being performed in conditions equivalent to a lower strain rate (although through imposed load conditions) compared to Batto and Schulson (1993), the stress concentration at the crack tips is not relaxed by the formation

10   of wing cracks but by plasticity mechanisms in the creep zone at the tip. Dislocations are therefore expected to nucleate and propagate at crack tips as shown by Rice and Thomson (1974). Recently, Argon (2001) showed that both nucleation of dislocations at crack tip, and the mobility of the nucleated dislocations come into play to induce the stress relaxation responsible for a crack arrest. Considering the high temperature conditions of our experiments, the dislocation multiplication leads to dynamic recrystallization mechanisms to occur in the creep zone nearby crack tips. Indeed, nucleation of new grains is observed very

15   close to the crack tips of crack 2, 3 and 4 (figure 5) and dislocation substructures as subgrains are being formed for instance around crack tips of cracks 1 and 2 (figure 9). These observations reveal that plasticity-driven recrystallization mechanisms are efficient to relax the local tensile stresses initiated at crack tips.

Local stresses associated with grain interactions during deformation of ice was indeed shown to be strongly heterogeneous, and to be responsible for the initiation of subgrain boundaries at the end of primary creep (Piazolo et al., 2015). Observation of crack initiation nearby grain boundaries and within grain interior is another evidence of such local stress concentration.

By following the strain field evolution all along the tests, we observe the closure of some parts of the cracks, in areas where nucleation and subgrain boundary formation were the most active. Crack closure is evidenced by the representation of principal strains which directions evolve from a tension component to a compressive component that ensures the recovering of continuity (figure 8d). In order to obtain a local closure of cracks, the stress field must provide a local compression component, perpendicular to the crack surface. The new microstructure formed by recrystallization mechanisms must therefore drive a redistribution of the local stress field to enable such a modification, still compatible with the macroscopic stress conditions.

Ductile fracture occurring at elevated temperature in metals can be related to void propagation, growth and coalescence. Recently, Shang et al. (2017) showed that DRX mechanisms were inducing a softening that reduces the local stress concentration, which serves as the driving force for this void-induced ductile fracture. Similar observations of a ductile to brittle transition in Olivine driven by plasticity mechanisms was deeply studied by Druiventak et al. (2011). In samples deformed at 20°C, 300°C and 600°C they observed microcracking at grain boundaries and in the grain interiors, but also arrays of dislocations related to crystal plasticity. Similarly to our observations, at the highest temperature, plasticity took place in the form of strongly misoriented undulatory extinctions (associated with various types of dislocations), deformation lamellae, and 3D dislocation cells inducing strong modifications of the microstructure. Our results therefore present some interest beyond the ice community. Similar procedures could very interestingly be applied to a wide range of materials in order to estimate the role of the level of plastic anisotropy on strain localization and on the efficiency of plasticity-driven recrystallization mechanisms to relax the local stress field at crack tips.

On top of the mechanical meaning of these observations, we highlight the fact that, since we were able to follow local crack closure during the test, care must be taken when performing experimental tests in conditions close to the ductile to brittle transition (typically at strain rates above $10^{-6}s^{-1}$, or compressive stress above 0.9 MPa), and at high temperature. Micro-cracking and DRX mechanisms can influence the local stress relaxation, and therefore the mechanical response, without leaving any track in the final microstructure.

**5 Concluding remarks**

The present work reveals, for the first time in ice, the evolution of the heterogeneous strain field during the onset of tertiary creep, in conditions where local cracking occurs to relax the local stress field. This observation was made possible by taking advantage of samples with 2D-1/2 microstructures from which surface observations follow bulk behavior.

While strain field localizes into bands with a length larger than the grain dimensions, cracks appear to relax stress concentration at the side of the strain localization bands, where deformation by dislocation glide must have been impeded by low local Schmid factor conditions.

Relaxation of the local stress field by crack opening results in a local redistribution of the strain field, as evidenced by the abrupt

weakening of some deformation bands after cracking. At the crack tips, where stress concentrates, plasticity-driven dynamic recrystallization mechanisms are observed as new small grains and high angle subgrain boundaries in the final microstructure. The new formed boundaries also appear visible on strain field patterns during the test, as new strain concentration areas. While induced by local stress concentration at crack tip, recrystallization mechanisms in turn generate a stress field redistribution as a result of microstructure modifications. This redistribution is indirectly evidenced by the modification of the measured strain field in the area, but also by the original observation of local crack closure, associated with a measured local compressive stress in place of the initial tensile stress responsible for the observed mode 1 crack opening. To conclude, the main results show that micro cracking and dynamic recrystallization mechanisms both resulting from a strongly heterogeneous stress field can coexist locally and that these mechanisms are efficient to relax local stresses at the ductile to brittle transition. Hence one may be careful when working at the frontiers of this transition since recrystallization can hide local cracking on the final microstructures.

*Author contributions.* T. Chauve, D. Georges and C. Lachaud performed the laboratory experiments. D. Georges and T. Chauve provided the data treatment. M. Montagnat and T. Chauve analysed the data and wrote the paper. P. Vacher provided some support for the DIC analyses and interpretation.

*Competing interests.* There is no competing interest

*Acknowledgements.* Financial support by the French "Agence Nationale de la Recherche" is acknowledged (project DREAM, ANR-13-BS09-0001-01). This work benefited from support from institutes INSIS and INSU of CNRS. It has been supported by a grant from Labex OSUG@2020 (ANR10 LABEX56) and from INP-Grenoble and UJF in the frame of a proposal called "Grenoble Innovation Recherche AGIR". Support from the imagery center IRIS of Grenoble-INP is acknowledged. MM benefited of a visitor research fellowship from WSL (Switzerland) in 2016-2017.

---

## Author Comment (AC2) · 20 Jun 2017

June 20th,

We are thankful for very interesting and helpful comments provided by both reviewers. We hope that we were able to make the best use of these comments to improve our paper. In black are the reviewer comments, and the authors response appear in between.

We provide a new version of the manuscript as a supplementary.

Referee #2

This manuscript describes original results on the DIC analyses of poly-crystalline ice

under creep deformation. As described in the authors' previous paper (Acta Materialia (2015)), application of the DIC method to ice provides a powerful tool to investigate evolution of strain fields during plastic deformation. In the present manuscript, very interesting results on behavior of local strain fields associated with cracking are presented, and the argument addressed are suitable for publication in Journal SE. However, I found some of the authors' explanations difficult to follow. The manuscript should be improved before acceptance for publication, with considering the following points:

(1) (General comment) For convincing argument, focusing on the experimental results found in the present study, distinguish more clearly those from others already presented in the previous papers by the author(s) and other researchers.

RESPONSE

Thanks for this comment. We tried to make this distinction by highlighting the main useful results from our previous studies in the introduction, p. 2 lines 15-19. Grennerat et al. 2012 made the first measurements of strain field evolution in ice polycrystals. Chauve et al. 2015 showed the evolution of local strain field with dynamic recrystallization processes (nucleation and subgrain boundary formation). Sentences were added p11, lines 13-16.

END

(2) Reconsider the description in 'Abstract'. The main purpose of the study must be to clarify 'the evolution of local strain fields around cracking' by the use of the DIC method as described in the top sentence in 'Abstract', but the description on the most important result obtained by the study is not clear. For example, if the argument is concluded by the last sentence 'A strong interaction between cracking and dynamic recrystallization is therefore evidenced', I wonder if it is a new finding. Such a general phenomenon may be already presented elsewhere. Consider carefully what is the most important finding made by the study. In addition, the title suggests 'strain heterogeneity' for the main topic of the paper but no descriptions about it in 'Abstract' and 'Concluding remarks'.

RESPONSE

Following your advices, the title was modified in "Strain field evolution at the ductile to brittle transition; a case study on ice." We also modified the abstract and put in the notion of heterogeneities, the link between changes in microstructures and stress redistribution. We left and modified the sentence rising the role of DRX as a way to relax the crack-tip stress field, since we didn't find any paper making such a direct relation. Although it might be specific to ice, we add "ice" into the sentence. Then, we tried to give more consistency between the conclusion and the abstract.

END

(3) 'Introduction' should be more concise, with focusing on the main topic of the paper.

RESPONSE

OK, we have shortened it a bit putting the accent to the ice case. We left the description of ice deformation behavior, and on previous work characterizing strain heterogeneities and evolution, in order to show that these studies were only performed in the ductile regime.

END

(4) I found very interesting results are presented in section 3 'Strain field evolution ....'. It should be emphasized more clearly what is found in the present study, and describe it also in 'Concluding remarks'.

RESPONSE

It is not clear for us which of the interesting results were not highlighted in the concluding remarks. - we mentioned the fact that cracks appear at the side of deformation bands, where we can expect higher Schmid factor conditions - we mentioned that cracking modifies the deformation band pattern - we mentioned the co-existence between micro-cracking and dynamic recrystallization mechanisms, with DRX plasticitydriven at the crack tips. - we therefore highlighted the efficiency of DRX mechanisms to relax local stresses. - and also the fact that changes in microstructure due to cracking and DRX induces a redistribution of local stress field... Maybe the reviewer could help by providing a clearer statement of what we missed? In the new version, we tried to improve the concluding remark part.

END

(5) (Line 5 to 6 on p.15. In section 5 'Concluding remarks') What does 'large' mean in 'large bands' (large in width, length, or thickness) ?

RESPONSE

'large bands' mean band with a length larger than the grain size. We added this information.

END

What is the difference between the 'band' in 'strain field localises into large bands' and the 'zone' in 'strain localization zones'?

RESPONSE

There is no difference between strain 'zones' and 'bands'. We clarified this in the text.

END

In addition, the description 'cracks appear nearby but not on the strain localization zones, where deformation by dislocation glide must have been impeded by low Schmidt factor conditions' is difficult to understand. If dislocation glide is impeded in the 'strain localisation zones', how does strain localise into the 'zones' ? A clear-cut description is required in 'Concluding remarks'.

RESPONSE

We meant that stress must concentrate at the side of strain localization bands, where

crack later appear to relax it, and one of the reason for such a stress concentration might be that there exist regions where local dislocation glide is easy (where strain concentrate), but if the region just nearby has a low Schmid factor, deformation can not occur, and stress concentrate up to cracking. This is why "nearby" is very important. We reformulated this part to make it clearer.

END

(6) (Line 9 to 10 on p.15) The description 'a strong redistribution of the local strain field such as already observed by Chauve et al. (2015)' should be revised to distinguish more clearly the original results obtained by the present study from the results already presented in other paper to avoid readers' misunderstanding in evaluation of this paper.

RESPONSE

This clarification has been done in the discussion part. We hope that it answers this comment.

END

(7) (Line 11 to 14 on p.15) This paragraph is not easy to follow because the experimental results (facts) and speculative descriptions are not well distinguished. As a concluding remark, what was found in the present study should be more clearly described.

RESPONSE

We tried to emphasize more clearly the main result of our study by reformulating this part.

END

Please also note the supplement to this comment:
http://www.solid-earth-discuss.net/se-2017-24/se-2017-24-AC2-supplement.pdf